# ALC1/eIF4A1-mediated regulation of *CtIP* mRNA stability controls DNA end resection

**Fernando Mejías-Navarro**[1,2☯], **Guillermo Rodríguez-Real**[1,2☯], **Javier Ramón**[1,2¤], **Rosa Camarillo**[1,2], **Pablo Huertas**[1,2]*

**1** Department of Genetics, University of Seville, Sevilla, Spain, **2** Centro Andaluz de Biología Molecular y Medicina Regenerativa-CABIMER, Universidad de Sevilla-CSIC-Universidad Pablo de Olavide, Sevilla, Spain

☯ These authors contributed equally to this work.
¤ Current address: Mitochondrial Disorders Unit, Vall d'Hebron Institut de Recerca, Barcelona, Catalonia, Spain
* pablo.huertas@cabimer.es

**Data Availability Statement:** All relevant data are within the manuscript and its Supporting Information files.

## Abstract

During repair of DNA double-strand breaks, resection of DNA ends influences how these lesions will be repaired. If resection is activated, the break will be channeled through homologous recombination; if not, it will be simply ligated using the non-homologous end-joining machinery. Regulation of resection relies greatly on modulating CtIP, which can be done by modifying: i) its interaction partners, ii) its post-translational modifications, or iii) its cellular levels, by regulating transcription, splicing and/or protein stability/degradation. Here, we have analyzed the role of ALC1, a chromatin remodeler previously described as an integral part of the DNA damage response, in resection. Strikingly, we found that ALC1 affects resection independently of chromatin remodeling activity or its ability to bind damaged chromatin. In fact, it cooperates with the RNA-helicase eIF4A1 to help stabilize the most abundant splicing form of *CtIP* mRNA. This function relies on the presence of a specific RNA sequence in the 5′ UTR of *CtIP*. Therefore, we describe an additional layer of regulation of CtIP—at the level of mRNA stability through ALC1 and eIF4A1.

## Author summary

The DNA molecule is constantly threatened by the appearance of physical or chemical modifications that endanger the integrity of the genetic information. Among them, the breakage of the DNA molecule is the most challenging to deal with. In order to minimize such risks, cells have developed multiple DNA repair mechanisms that take care of broken chromosomes. The regulation of the usage of each different mechanism is extremely important for the safekeeping of the genetic material. CtIP is a key protein of pivotal importance in the decision between different broken DNA repair pathways. Here we show a novel regulatory mechanism that controls the abundance of this protein by controlling the stability of the different mRNA molecules produced by the *CtIP* gene. In brief, we show that the factors ALC1 and eIF4A1 affect the stability of some, but not all, *CtIP*

**Funding:** This work was financed by an R+D+I grant from the Spanish Ministry of Economy and Competitivity (SAF2016-74855-P) and by the European Union Regional Funds (FEDER). FM-N was funded with an FPU fellowship from the Spanish Ministry of Education, and JR and GR-R were supported by the Regional Government of Andalucía (Junta de Andalucía) with a contract of the program "GARANTÍA JUVENIL EN LA UNIVERSIDAD DE SEVILLA". CABIMER is supported by the regional government of Andalucía (Junta de Andalucía). The funders had no role in study design, data collection and analysis, decision to publish, or preparation of the manuscript.

**Competing interests:** The authors have declared that no competing interests exist.

mRNA variants. This effect is dependent on the presence or absence of a specific RNA structure, called a G-quartet or G-quadruplex, in a non-coding portion of the mRNA.

## Introduction

In order to maintain genomic integrity, cells have to successfully deal with thousands of threats on a daily basis that could potentially compromise the structure or sequence of DNA [1]. While DNA double-strand breaks (DSBs) occur at a low incidence, they, nonetheless, are very challenging to repair faithfully [1]. The appearance of one or more DSBs triggers a very complex response that is required to minimize genomic instability. This includes the activation of specific DSB repair pathways as well as of a cellular response that will affect virtually every aspect of metabolism, from cell cycle progression, to gene and protein expression [1]. In terms of pure repair, broken DNA molecules can be repaired by the error-prone, but relatively fast and simple, non-homologous end-joining (NHEJ) mechanism or by the more accurate but complex homologous recombination (HR) mechanism [2,3]. HR is considered the most error-free way to deal with a broken chromosome, at least when the sister chromatid is present [3].

HR is a multi-step and complex pathway, but it is always initiated by processing the DNA ends in an event known as DNA end resection [4,5]. Resection consists of a 5′–3′ degradation of one DNA strand at each side of the break, forming long tails of 3′-OH single-strand DNA (ssDNA). ssDNA must be protected from nucleases by the binding of protective protein complexes, such as RPA [4,5]. Importantly, ssDNA is an obligatory intermediate of all HR pathways and is also a potent inhibitor of NHEJ. Resection itself is a two-step mechanism. Its initiation is catalyzed by the action of the Mre11-Rad50-Nbs1 (MRN) complex [4,5]. However, several cellular and local criteria must be met for resection to be activated; therefore, resection is only started at specific breaks under specific circumstances.

Resection licensing depends greatly on a single factor, CtIP, which receives and integrates multiple cellular signals and is responsible for activating the MRN-role in resection [4,6]. Not surprisingly, CtIP is heavily regulated at several levels. First, the overall amount of CtIP is strictly controlled, both at the level of *CtIP* mRNA transcription [7–9] and by protein stability [10–12]. Second, several post-translational modifications (both constitutive and induced) of CtIP are required for it to be able to help activate resection [11–17]. Once resection is initiated by the combined actions of MRN and CtIP, it is extended by other nucleases [5,18], and the break can be repaired by recombination. Finally, a plethora of anti- and pro-resection factors can exert a regulatory role at this level [5].

In addition to the activation of the repair mechanisms, the presence of DSBs triggers a complex, massive cellular response known as the DNA damage response (DDR) [1]. This consists of a signaling cascade that can alter all cellular metabolism (including cell cycle progression, mRNA expression and splicing, and the chromatin environment) to create the perfect environment for repair to take place [19]. In many cases, there is a crosstalk between the DNA repair pathways and DDR. Indeed, many repair factors are controlled at different levels by this global response, either through changes in their transcription or protein stability, or through post-translational modifications.

Thousands of proteins are involved in DDR, many of which can be recruited to regions of chromatin flanking a DSB. One such protein is ALC1, a chromatin remodeling factor that is recruited very rapidly to damaged DNA and is involved in local decondensation of chromatin, which is known to facilitate DDR [20]. Also known as CHD1L (for CHD1-like), ALC1 lacks the chromodomains typical of the members of the CHD family but maintains the helicase

domain [20]. Additionally, ALC1 has a macrodomain that is involved in PAR-dependent recruitment to DNA damage [20]. Further, ALC1 can locally relax the chromatin surrounding DSBs [21]. In addition to its role in DDR, ALC1 has also been implicated in transcription of specific genes [22,23]. Not surprisingly, ALC1 is considered an oncogene with effects in promoting proliferation, cell migration, invasion, and metastasis, and in inhibiting apoptosis [24]. Despite its influence on DDR and oncogenesis, ALC1 has not yet been shown to be involved directly in DNA repair.

Here we show that ALC1 indeed facilitates DNA end resection and homologous recombination and, as a consequence, impairs NHEJ. Strikingly, however, it does so in a fashion that is completely independent of its DDR function, its helicase activity, and its PAR-dependent binding to damaged chromatin. Rather, this role is exerted by controlling CtIP expression and relies on the presence of a G-quadruplex (G4) structure in the 5′-UTR of the *CtIP* major mRNA isoform. Remarkably, whereas the G4-containing (G4) *CtIP* mRNA is less stable in the absence of ALC1, a second mRNA species that lacks such a structure in the 5′-UTR (G4less) is not. The mRNA helicase eIF4A1, which is involved in unwinding G4 in the 5′-UTR of mRNAs, cooperates with ALC1 in this role. In fact, the G4 5′-UTR sequence of *CtIP* is sufficient to affect expression of the non-related control *GFP* gene in a way that depends on both eIF4A1 and ALC1. Thus, we have revealed an additional way in which ALC1 and eIF4A1 regulate the responses to DSBs—by directly controlling *CtIP* mRNA stability.

## Materials and methods

### Cell lines and growth conditions

U2OS, HeLa and RPE1 human cell lines were grown in DMEM (Sigma-Aldrich) supplemented with 10% fetal bovine serum (Sigma-Aldrich), 2 mM L-glutamine (Sigma-Aldrich) and 100 units/ml penicillin and 100 μg/ml streptomycin (Sigma-Aldrich). U2OS expressing YFP-ALC1 variants [20] were grown in standard U2OS medium supplemented with 0.5 mg/ml G418 (Gibco, Invitrogen). U2OS cells stably expressing a control shRNA or an shRNA against CCAR2 or 53BP1, and those cells bearing a copy of the DR-GFP, SA-GFP or EJ5-GFP reporter system, were grown in standard DMEM described above supplemented with 1 μg/ml puromycin (Sigma-Aldrich). U2OS cells stably expressing the GFP or GFP-CtIP plasmid were grown in standard DMEM supplemented with 0.5 mg/ml G418 (Sigma-Aldrich). For serum starvation and release, RPE1 cells were cultured for 24h in medium containing 0.1% FBS and then released in fresh medium complemented with 10% FBS.

### siRNAs, plasmids and transfections

siRNA duplexes were obtained from Sigma-Aldrich, Dharmacon or Qiagen (S1 Table) and were transfected using RNAiMax Lipofectamine Reagent Mix (Life Technologies), according to the manufacturer's instructions. Plasmid transfection of U2OS cells with YFP-ALC1 variants [20] and GFP plasmids containing the 5′ UTR regions of CtIP was carried out using FuGENE 6 Transfection Reagent (Promega) according to the manufacturer's protocol.

### HR and NHEJ analysis

U2OS cells bearing a single copy integration of the reporters DR-GFP (Gene conversion), SA-GFP (SSA) or EJ5-GFP (NHEJ) [25,26] were used to analyze the different DSB repair pathways. In all cases, 35,000 cells were plated in 6-well plates in duplicate. One day after seeding, cells were transfected with the indicated siRNA, and medium was replaced with fresh one without siRNAs 6–8 h later. Two days after the start of siRNA transfection, each duplicate

culture was infected, one well with lentiviral particles containing I-SceI–BFP expression construct at MOI 10 using 4 μg/ml polybrene in 1 ml of DMEM, and the other well with DMEM including polybrene but without lentivirus (as a control for basal fluorescence). Cells were then left to grow for an additional 24 h before changing the medium for fresh DMEM. One day later, cells were washed with PBS, trypsinized, neutralized with DMEM, centrifuged for 5 min at 700 g, fixed with 4% paraformaldehyde for 20 min and collected by centrifugation. Cell pellets were washed once with PBS before resuspension in 150 μl of PBS. Samples were analyzed with a BD FACSAria with the BD FACSDiva Software v5.0.3. Four different parameters were considered: side scatter (SSC), forward scatter (FSC), blue fluorescence (407 nm violet laser BP, filter 450/40), green fluorescence (488 nm blue laser BP, filter 530/30). Finally, the number of green cells from at least 10,000 events positives for blue fluorescence (infected with the I-SceI–BFP construct) was scored, considering the background of green fluorescence obtained in the samples without infection with lentivirus harboring pBFP-ISceI plasmid as previously described [26–28]. To facilitate comparison between experiments, this ratio was normalized with siRNA control. At least three independent experiments were carried out for each condition; the average and standard deviations are given.

## SDS-PAGE and Western blot analysis

Protein extracts were prepared in 2× Laemmli buffer (4% SDS, 20% glycerol, 125 mM Tris-HCl, pH 6.8) and passed 10 times through a 0.5 mm needle–mounted syringe to reduce viscosity. Proteins were resolved by SDS-PAGE and transferred to low fluorescence PVDF membranes (Immobilon-FL, Millipore). Membranes were blocked with Odyssey Blocking Buffer (LI-COR) and blotted with the appropriate primary antibody (S2 Table) and infra-red dyed secondary antibodies (LI-COR) (S3 Table). Antibodies were prepared in blocking buffer supplemented with 0.1% Tween-20. Membranes were air-dried in the dark and scanned in an Odyssey Infrared Imaging System (LI-COR), and images were analyzed with ImageStudio software (LI-COR).

## YFP-ALC1 immunoprecipitation

To immunoprecipitate YFP-ALC1 protein, U2OS cells were scrapped in lysis buffer containing proteases and phosphatases inhibitors (50 mM Tris-HCl pH 7.5, 50 mM NaCl, 1 mM EDTA, 0.5% NP-40, protease inhibitors [11873580001, Roche] and phosphatase inhibitor cocktail 3 [P0044, Sigma]). To degrade DNA, 100 U/ml Benzonase (70746–4, VWR) was added to protein extracts and incubated 30 min on ice. During that time, chromatin was sheared by passing the sample three times through a syringe with a 0.5x16 mm needle. In parallel, GFP-Trap beads (gtm-20, chromotek) were equilibrated once in lysis buffer without inhibitors for 5 min on ice with rocking, followed by two washes with lysis buffer containing inhibitors. Then, 1 mg of protein extracts was diluted in lysis buffer containing inhibitors to 0.2% NP-40 concentration. GFP-Trap beads were added and incubated under gentle agitation at 4˚C for 2 hours. Beads were then washed twice with lysis buffer containing inhibitors, and the precipitate was eluted in Laemmli buffer, boiled, and resolved in SDS-PAGE as described.

## Immunofluorescence and microscopy

For RPA foci visualization, U2OS cells knocked-down for different proteins were seeded on coverslips. At 1 h after either irradiation (10 Gy) or treatment with 10 μM etoposide, coverslips were washed once with PBS followed by treatment with pre-extraction buffer (25 mM Tris-HCl, pH 7.5, 50 mM NaCl, 1 mM EDTA, 3 mM MgCl$_2$, 300 mM sucrose and 0.2% Triton X-100) for 5 min on ice. Cells were fixed with 4% paraformaldehyde (w/v) in PBS for 15 min.

Following two washes with PBS, cells were blocked for 1 h with 5% FBS in PBS, co-stained with the appropriate primary antibodies (S2 Table) in blocking solution overnight at 4˚C or for 2 h at room temperature, washed again with PBS and then co-immunostained with the appropriate secondary antibodies for 1 h (S3 Table) in blocking buffer. After washing with PBS and drying with ethanol 70% and 100% washes, coverslips were mounted into glass slides using Vectashield mounting medium with DAPI (Vector Laboratories). RPA foci immunofluorescences were analyzed using a Leica Fluorescence microscope.

For 53BP1 visualization, U2OS cells were seeded and transfected as previous described. Once collected, cells were fixed with methanol (VWR) for 10 min on ice, followed by treatment with acetone (Sigma) for 30 sec on ice. For RIF1 foci visualization, cells were fixed with 4% PFA for 15 min, washed twice with 1× PBS and then permeabilized for 15 min with 0.25% Triton diluted in 1× PBS. Finally, CCAR2 foci were observed in cells permeabilized for 10 min on ice with 0.2% Triton diluted in 1× PBS.

Samples were immunostained as described above with the appropriate primary and secondary antibodies (S2 and S3 Tables). Images obtained with a Leica Fluorescence microscope were then analyzed using Metamorph to count the number, intensity and size of the foci.

## SMART (single-molecule analysis of resection tracks)

SMART was performed as described [29]. Briefly, cells were grown in the presence of 10 μM BrdU for 22–24 h. Cultures were then irradiated (10 Gy) and harvested after 1 h. Cells were embedded in low-melting agarose (Bio-Rad), followed by DNA extraction. DNA fibers were stretched on silanized coverslips, and immunofluorescence was carried out to detect BrdU (S2 and S3 Tables). Samples were observed with a Nikon NI-E microscope, and images taken and processed with the NIS ELEMENTS Nikon Software. For each experiment, at least 200 DNA fibers were analyzed, and the length of the fibers was measured with Adobe Photoshop CS4.

## Cell cycle analysis

Cells were fixed with cold 70% ethanol overnight, incubated with 250 μg/ml RNase A (Sigma) and 10 μg/ml propidium iodide (Fluka) at 37˚C for 30 min and analyzed with a FACSCalibur (BD). Cell cycle distribution data were further analyzed using ModFit LT 3.0 software (Verity Software House Inc).

## RNA extraction, reverse transcription and quantitative PCR

RNA extracts were obtained from cells using NZY Total RNA Isolation kit (Nzytech) according to manufacturer's instructions. To obtain complementary DNA (cDNA), 1 μg RNA was subjected to RQ1 DNase treatment (Promega) prior to reverse transcription reaction using Maxima H Minus First Strand cDNA Synthesis kit (Thermo Scientific) according to manufacturer's instructions. Quantitative PCR from cDNA was performed to check siRNA-mediated knock-down of several proteins. For this, iTaq Universal SYBR Green Supermix (Bio-Rad) was used following manufacturer's instructions. DNA primers used for qPCR are listed in S4 Table. Q-PCR was performed in an Applied Biosystem 7500 FAST Real-Time PCR system. The comparative threshold cycle (Ct) method was used to determine relative transcripts levels (Bulletin 5279, Real-Time PCR Applications Guide, Bio-Rad), using β-actin expression as internal control. Expression levels relative to β-actin were determined with the formula $2^{-\Delta\Delta Ct}$ [30].

### Cycloheximide chase assay

U2OS cells depleted for the indicated siRNA were treated with fresh DMEM containing 150 μg/ml cycloheximide (CHX, Sigma-Aldrich). Cells were collected 0, 4 and 8 h after CHX addition, and protein extracts were prepared as described before. Samples were resolved by SDS-PAGE, analyzed by immunoblotting (S2 and S3 Tables) and quantified with ImageStudio software (LI-COR). The siNT condition at time 0 was normalized as 100%, and the other conditions were relativized to this point. CtIP expression was calculated as the amount of CtIP divided by α-tubulin signal.

### Cloning of *CtIP* 5′-UTR sequences

Total RNA from U2OS cells was extracted and reverse-transcribed into cDNA as previous described using an specific primer for *CtIP* mRNA (S4 Table). The 5′-UTR regions corresponding to the G4less and G4 isoforms were amplified by PCR by adding specific target sites for restriction enzymes (Takara). Amplified regions were cloned into the pEGFP-N1 vector (Clontech). Plasmid DNA was inserted into competent cells of the DH5α strain of *Escherichia coli*. Vector DNA was purified using PureYield Plasmid Maxiprep System (Promega) and sequenced to check the proper integration of the insert into the GFP plasmid.

### Statistical analysis

Statistical significance was determined with a Student's *t*-test or ANOVA as indicated using PRISM software (Graphpad Software Inc.). Statistically significant differences were labelled with one, two or three asterisks for $P < 0.05$, $P < 0.01$ or $P < 0.001$, respectively.

## Results

### ALC1 facilitates homologous recombination by promoting resection

ALC1 is rapidly recruited to DSBs, suggesting a very early role for it in the response to such type of damage [20]. This observation raised the question whether ALC1 is affecting the choice between DSB repair pathways, namely between HR and NHEJ. To test this idea, we analyzed previously described, pathway-specific, GFP-based, DSB repair assays [25,26]. Strikingly, ALC1 depletion led to a global decrease in homologous recombination, both in Rad51-dependent gene conversion and in Rad51-independent single-strand annealing (Fig 1A and 1B). Reciprocally, downregulation of ALC1 increased NHEJ (Fig 1C). Similar results for homologous recombination were obtained using two additional siRNAs against ALC1, including one against the 3′-UTR (S1A Fig). However, NHEJ was more variable using these additional siRNAs (S1B Fig) (the levels of ALC1 depletion in U2OS cells with different siRNAs are shown in S1C and S1D Fig). Thus, our data agreed with ALC1 playing a role in the decision between HR and NHEJ at an initial level, and mostly at the level of recombination. Such unbalancing of DSB repair pathways has been observed mainly when DNA resection regulation is affected. In agreement with a role of ALC1 in DNA end processing, ALC1 depletion qualitatively resembled the downregulation of the key resection factor CtIP (Fig 1A–1C and S1A and S1B Fig).

Thus, one possibility was that ALC1 was affecting DNA end resection. To test this, we analyzed DNA end resection by checking the formation of RPA foci in cells exposed to DNA damage. We used phosphorylation of the histone variant H2AX (γH2AX) as a positive marker of DNA damage. Indeed, ALC1 depletion impaired resection in U2OS cells treated with ionizing radiation (Fig 2A) or the topoisomerase II inhibitor etoposide (Fig 2B). DNA end resection is regulated by the cell cycle, as G1 cells do not resect extensively [4]. We did not observe a significant change in the amount of G1 cells when ALC1 was downregulated (S1E Fig); however, to

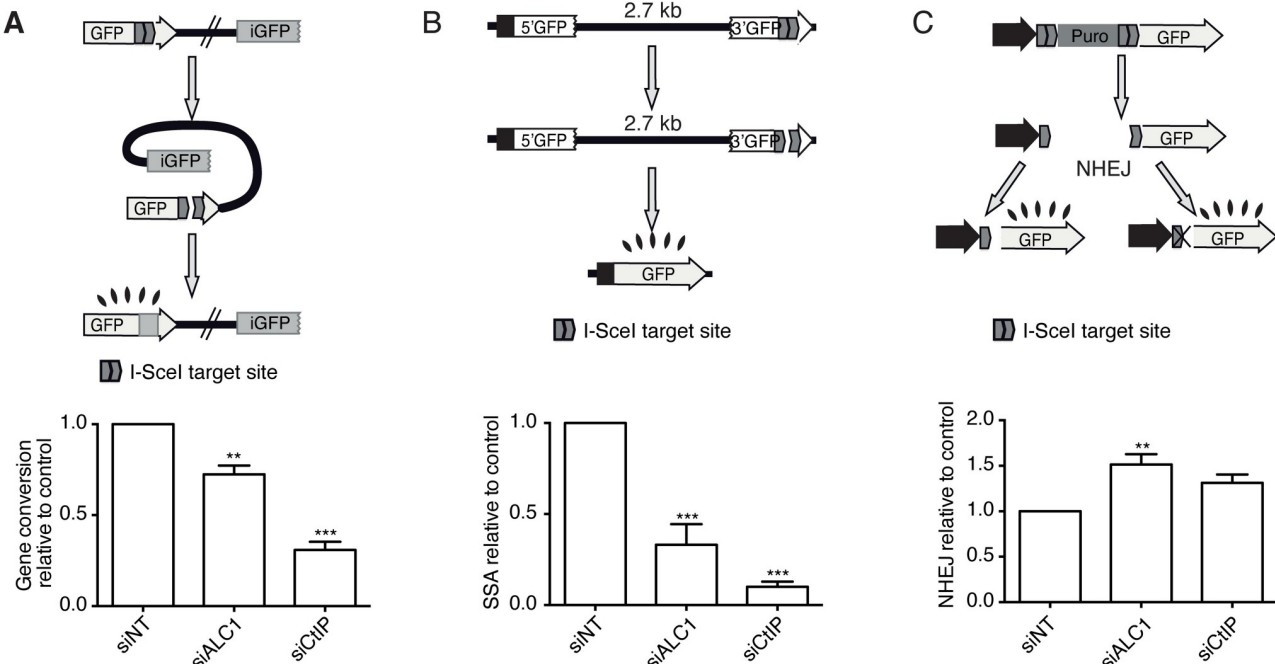

**Fig 1. ALC1 facilitates homologous recombination and impairs NHEJ. A** Classical homologous recombination was measured as described in the methods section using the DR-GFP reporter (top) [25]. An active GFP gene is formed upon gene conversion of an I-SceI–induced DSB. Briefly, cells constitutively transfected with a single copy of the reporter were depleted for the indicated proteins, and after two days were infected with viral particles containing an I-SceI–BFP expression construct. Two day later, fluorescence accumulation was scored by FACs. The percentage of blue fluorescent cells (bearing the I-SceI) that became green was scored and normalized as described in the Methods section. The experiment was repeated three times, and the average and standard deviation is plotted (bottom). Statistical significance was calculated using a one-way ANOVA test. Two or three asterisks represent $P < 0.01$ and $P < 0.001$, respectively. **B** Same as (A) but using the SSA reporter SA-GFP [26]. **C** Same as (A) but using the NHEJ reporter EJ5-GFP [26].

be absolutely sure that the observed effects were not due to cell cycle perturbations, we repeated the RPA foci assay after irradiation but considering only the cells positive for the cell cycle marker CENPF, a kinetochore protein that accumulates in S and G2 cells [31]. Even analyzing exclusively S/G2 cells, the DNA resection defect caused by ALC1 depletion was maintained (Fig 2C). Moreover, in HeLa cells, RPA foci formation after ionizing radiation was also compromised in cells downregulated for ALC1 (Fig 2D; see also S1C Fig for depletion, and S1F Fig for cell cycle). From this point onwards, unless specified, all the experiments shown were performed in U2OS cells.

To demonstrate that the effect in DNA end resection was indeed mediated by ALC1, we complemented the depletion of the mRNA using a siRNA against the 3′-UTR with a siRNA-resistant YFP-ALC1 construct (Fig 2E). We observed a reduction not only in the number of breaks that were processed in ALC1-depleted cells but also in the length of the resected DNA, measured by SMART [29,32] (Fig 2F). Moreover, we observed a decrease after ALC1 depletion not only in the number of RPA foci–positive cells, but also in the average number of RPA foci per cell (S1G Fig).

## ALC1 depletion increases the loading of anti-resection factors

For resection to proceed, it needs to overcome the barriers caused by several anti-resection pro-NHEJ proteins that are recruited to damaged chromatin, such as 53BP1, RIF1 and CCAR2 [33,34]. Thus, we wondered if the impaired resection observed in ALC1-depleted cells

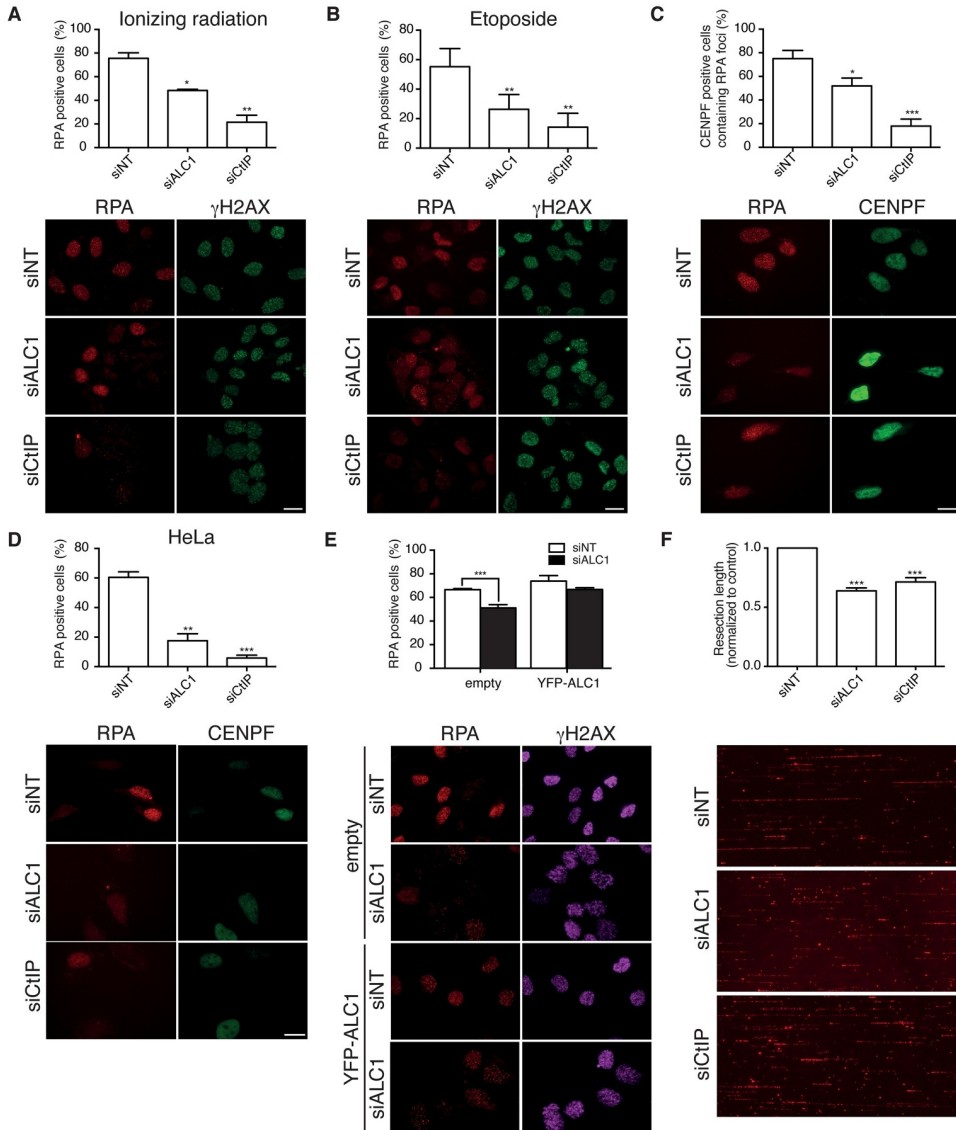

**Fig 2. ALC1 is required for DNA end resection. A** DNA end resection was scored by the accumulation of RPA foci 1 h after 10 Gy of ionizing radiation in U2OS cells depleted for two days for the indicted factors. γH2AX was used as a mark of DNA damage. The average and standard deviation of three independent experiments is plotted (top), and representative images are shown (bottom). Scale Bar represent 25 μm. Statistical significance was calculated using a one-way ANOVA test. One, two or three asterisks represent P<0.05, P<0.01 and P<0.001 respectively. **B** Same as (A) but after 1 h treatment of 10 μM etoposide. **C** Same as (A) but using CENPF as a marker of the S and G2 phases of the cell cycle. The percentage of RPA-positive cells from those positive for CENPF is represented in the graph. **D** Same as (A) but in HeLa cells. **E** Same as (A) but in cells transfected or not with a YFP-tagged version of wild-type ALC1. In this case, an siRNA against the 3′-UTR of ALC1, absent in the YFP-fusion, was used to deplete endogenous ALC1. **F** Resection length on individual DNA fibers was calculated as previously published using SMART. Briefly, cells depleted for the indicated proteins were incubated for 24 h with BrdU. Cells were then irradiated with 10 Gy, and after 1 h, DNA was extracted as described in the Method section. DNA was then stretched on coverslips, and ssDNA was detected using a BrdU antibody. At least 250 DNA fibers were scored per condition. Resection length was normalized to the control sample. The average and standard deviation of three independent experiments were plotted (top). and representative images are shown (bottom). Statistical significance was calculated using a one-way ANOVA test. Three asterisks represent P<0.001.

correlated with an increased loading of such factors, as suggested by the increase in the NHEJ pathway (Fig 1C). Indeed, ALC1 downregulation, similar to CtIP depletion, increased the number of foci of both 53BP1 and RIF1 in irradiated cells (Fig 3A and 3B). This was not due to an accelerated kinetic of recruitment, as the levels of 53BP1 remained equally elevated at different times after irradiation (S2A Fig). Moreover, for 53BP1 (but not for RIF1), these foci were brighter and bigger (S2B–S2E Fig). Along the same lines, ALC1 depletion increased the number of foci of the resection antagonist CCAR2 (Fig 3C). Neither the size of CCAR2 foci nor their intensity were significantly different in siRNA-depleted ALC1 cells as compared to control cells (S2F–S2G Fig).

There are two possible explanations for the resection phenotype observed in ALC1 downregulated cells: 1) the chromatin environment favored either the recruitment or retention of anti-resection proteins, which in turn blocked processing of the DNA ends; and 2) the resection impairment caused an accumulation of 53BP1, RIF1 and CCAR2. In the first scenario, depletion of such resection barriers should suppress, or at least alleviate, the decrease in RPA foci caused by ALC1 reduction. As this was not observed (S3A–S3C Fig), we concluded that ALC1 facilitates resection progression directly, and that this resection is responsible for displacing 53BP1, RIF1 and CCAR2 from damaged DNA.

## The role of ALC1 in resection is independent of its catalytic activity and DNA binding

Several domains have been previously described in ALC1 as important for its role in the DNA damage response. This include two tandem catalytic domains with helicase activity, and a

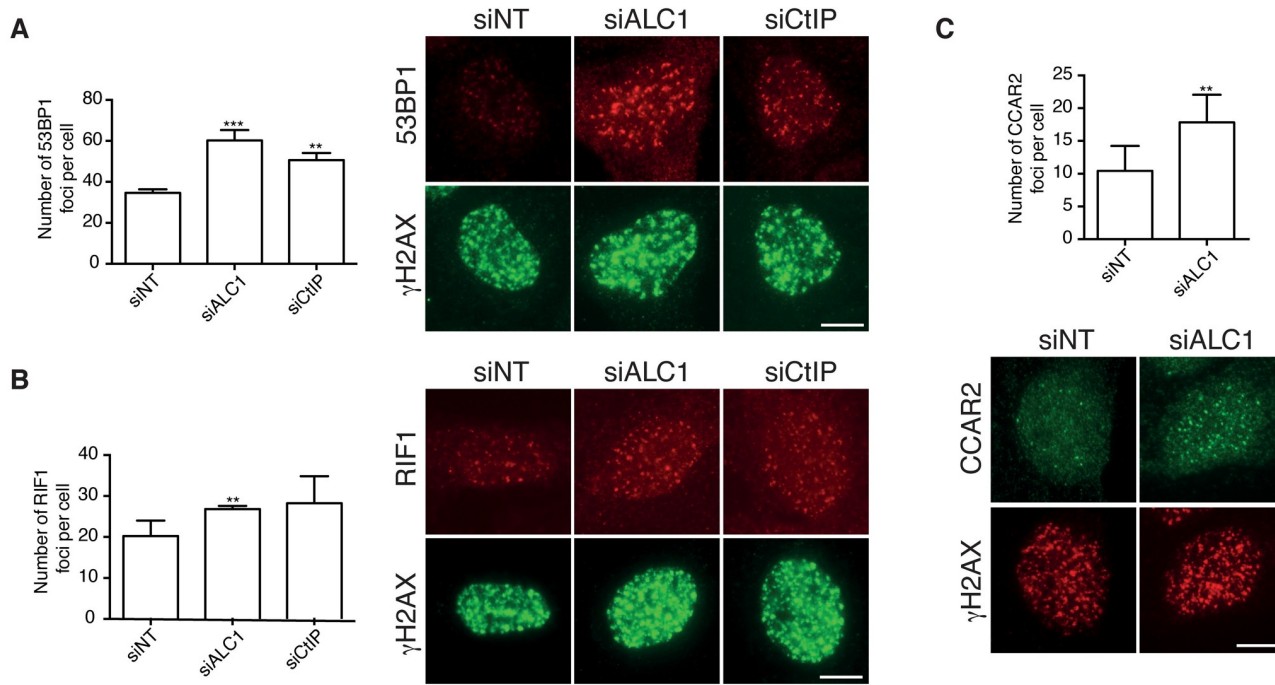

**Fig 3. The recruitment of anti-resection factors is increased upon ALC1 depletion. A–C** The formation of 53BP1 foci (A) or RIF1 foci (B), or the recruitment of CCAR2d (C) was tested in U2OS cells transfected with the indicated siRNAs at 1 h after irradiation (10 Gy). Cells were treated as described in Fig 2A. The number of foci per cell was scored automatically using the software Metamorph. The average and standard deviation of three independent experiments were plotted (left), and representative images of one cell are shown (right). Scale Bar represent 10 μm. Statistical significance was calculated using a one-way ANOVA test. Two or three asterisks represent P<0.01 and P<0.001, respectively.

macro-domain involved in PAR-dependent binding to damaged chromatin [20]. We tested if any of these factors are important for DNA resection. Surprisingly, ALC1-depleted cells expressing siRNA-resistant variants of the protein harboring mutations that compromise either the catalytic domain or the DNA binding domain [20] were as proficient in DNA resection as the wild-type protein, both with respect to the number of RPA-positive cells and to the average number of RPA foci per cell (S4A–S4D Fig). These data suggest that both domains are dispensable for the role of ALC1 in resection.

## ALC1 controls CtIP protein levels

Our data suggested that ALC1 controlled resection directly, but in a manner that required neither its DNA binding capacity nor its helicase activity. Thus, we considered the possibility that it was somehow affecting some factors from the resection machinery. Due to its key role in DNA end resection, we first focused on CtIP. Indeed, a reduction in CtIP levels was readily observed after ALC1 depletion (S1C and S1D Fig). We then performed a cycloheximide chase to analyze the stability of CtIP after ALC1 depletion (Fig 4A and 4B). Indeed, ALC1-depleted cells showed reduced levels of the key resection factor CtIP (Fig 4A; see time 0), which was however not due to a reduction of protein stability (Fig 4A and 4B). Interestingly, these decreased protein levels were rescued by expression of wild-type YFP-ALC1 as well as by the mutants defective in the catalytic activity and DNA damage recruitment (S4E Fig), in agreement with these domains as being dispensable for the resection phenotype of this factor. ALC1, as a chromatin remodeling factor, has been involved in transcription [22,23], so a reasonable hypothesis was that ALC1 affected the transcription of CtIP. However, both the catalytic activity and the DNA binding domains of ALC1 were dispensable for DNA end resection and protein levels (S4A–S4D Fig). Indeed, total *CtIP* mRNA levels were only slightly reduced after downregulation of ALC1 (Fig 4C). Thus, our data suggested that ALC1 controlled CtIP abundance post-transcriptionally but pre-translationally.

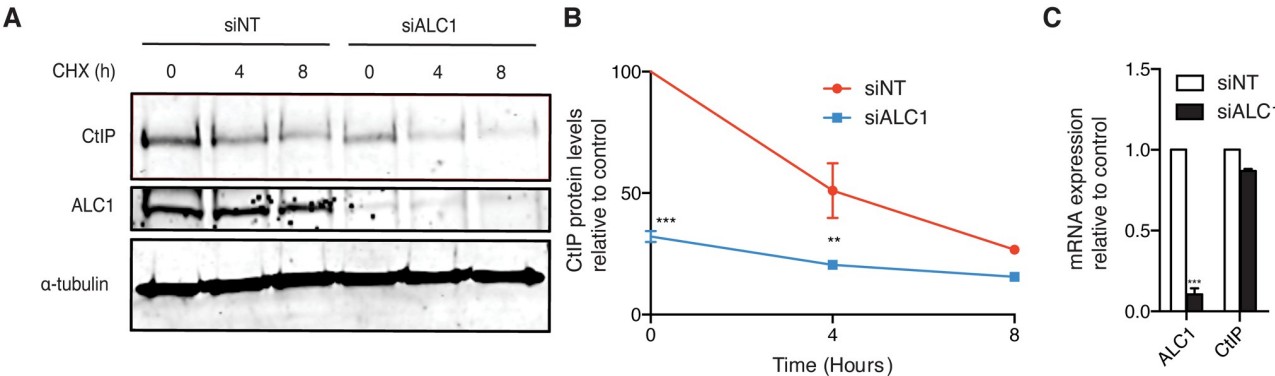

**Fig 4. CtIP protein levels are reduced in cells downregulated for ALC1. A** U2OS cells transfected with the indicated siRNA were treated with cycloheximide (CHX). Protein samples were collected 0, 4 and 8 h after 150 μg/ml CHX addition to the cell culture, resolved in SDS page and blotted for CtIP, ALC1 and tubulin (as a loading control). A representative western blot (of three) is shown. **B** Quantification of (A). The average and standard deviations of three independent experiments are shown. Statistical significance of the differences at each time point were calculated with a t-student test and are marked with two ($P < 0.01$) or three ($P < 0.001$) asterisks. **C** Total RNA was collected from U2OS cells transfected with siRNA against ALC1 (black bars) or a control sequence (white bars). The amount of ALC1 and *CtIP* mRNA was then quantified by qPCR using primers that recognize all splicing variants of the gene. Values in ALC1-depleted cells were normalized to control samples. The average and standard deviations of three independent experiments are shown. Statistical significance was calculated using a two-way ANOVA test. Three asterisks represent P<0.001.

## ALC1 regulates CtIP in coordination with the translation factor eIF4A1

ALC1 was previously found to physically interact with the eIF4A1 translation factor in a mass spectrometry experiment [20]. eIF4A1 is a RNA helicase involved in opening RNA G-quadruplexes (G4s) in long 5′-UTRs, which allows their translation [35]. We thus wondered if the regulatory function of ALC1 over *CtIP* mRNA in the post-transcription–pre-translation window was dependent on its association with eIF4A1. We first co-immunoprecipitated eIF4A1 with YFP-ALC1 using a GFP-trap to confirm the interaction in our system (Fig 5A). Despite some background unspecific binding, we observed an increase of eIF4A1 signal when the YFP-ALC1 was used as a bait (Fig 5A). Intriguingly, we also observed that eIF4A1 siRNA-mediated depletion also decreased CtIP levels to a similar extent as ALC1 (Fig 5B and S5A Fig). In fact, double depletion of both proteins showed a similar phenotype to either single depletion, arguing that they could belong to the same genetic pathway (Fig 5B and S5A Fig). Similarly, eIF4A1 downregulation mimicked ALC1 depletion defect with respect to DNA end resection, measured as RPA foci formation after exposure to ionizing radiation (Fig 5C and 5D). (Note that, although eIF4A1 depletion changed the profile of the cell cycle (S5B Fig), this cannot account for the RPA defects). Cells simultaneously depleted for both ALC1 and eIF4A1 behaved very similarly to those transfected with either siRNA independently in terms of resection (Fig 5C), reinforcing the idea that there is a genetic connection between them. Moreover, eIF4A1 depletion impaired homologous recombination, measured using the DR-GFP reporter, and its effect was epistatic over ALC1 depletion (Fig 5E). These results strongly suggest that ALC1 and eIF4A1 collaborate in resection and recombination.

One prediction of our model of ALC1 and eIF4A1 cooperating to maintain CtIP levels is that overexpression of CtIP should rescue, or at least alleviate, the resection phenotype observed when either of these two proteins is downregulated. Indeed, ectopic expression of CtIP cDNA, lacking the endogenous 5′-UTR, fused to GFP (GFP-CtIP; Fig 5A) completely rescued RPA foci formation in cells depleted of ALC1, eIF4A1 or both (Fig 5B and 5C). Indeed, upon GFP-CtIP overexpression, resection in ALC1- and eIF4A1-downregulated cells could be rescued to by GFP-CTIP to a similar extent as CtIP depletion. Interestingly, only the resection defect of ALC1, but not the HR impairment, can directly be ascribed to the reduction of CtIP levels, as overexpression of siRNA resistant FLAG-CtIP did not suppressed such phenotype (S5C Fig). Note that FLAG-CtIP could not completely rescue CtIP depletion, arguing that the construct is not fully functional but enough to overcome CtIP defect. Thus, this suggests that ALC1 role in resection is mediated by CtIP, but that ALC1 plays additional roles in recombination at later steps that are independent of CtIP levels.

## ALC1 and eIF4A1 affect the stability of specific *CtIP* mRNA species

Our data suggested that eIF4A1 and ALC1 act together to control CtIP levels in a post-transcriptional–pre-translational window, suggesting that ALC1 has a yet- uncharacterized function in mRNA metabolism. eIF4A1 is particularly important for translate mRNAs containing a (CGG)4 structure [35]. Strikingly, *CtIP* gene produces two different mRNAs molecules that code for the canonical full-length protein and vary only in their 5′-UTR sequences. Using the software QGRS Mapper, we found that one of those transcripts has a sequence prone to form G4s (Fig 6A, green boxes of "G4" transcript). On the contrary, the other variant lacks such structure (Fig 6A; "G4less" transcript). Further, although both transcripts have several (CGG)4 motifs that are known to be recognized by eIF4A1 [35,36], the G4 variant has more of them (Fig 6A, marked in red).

We decided to use specific sets of primers to test the accumulation of each of those two mRNA in cells by quantitative PCR. Of note, the CtIP splicing variants that do not code for the

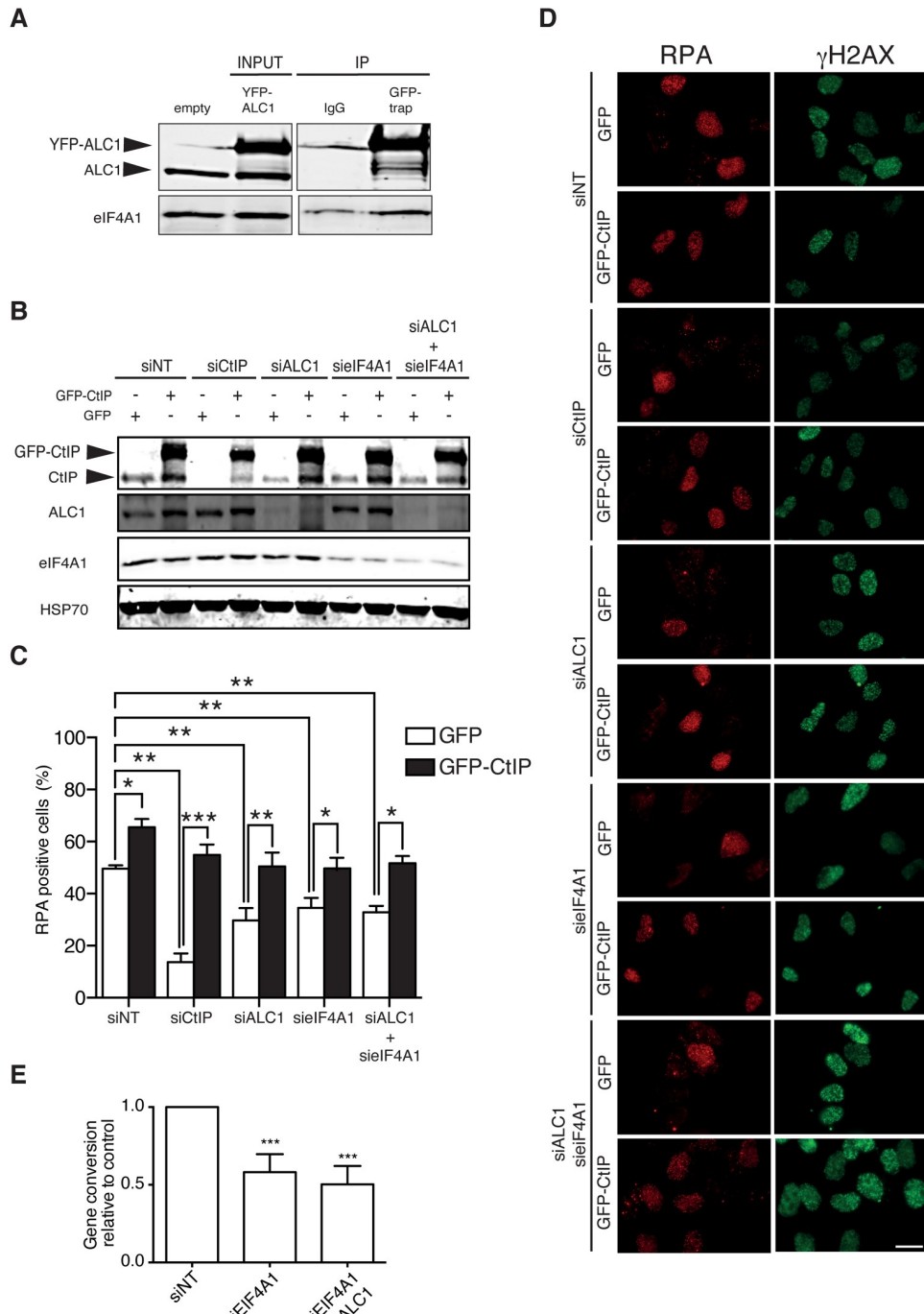

**Fig 5. eIF4A1 and ALC1 affect resection by controlling CtIP levels. A** Interaction between ALC1 and eIF4A1 was detected by immunoprecipitation (IP, left) of protein samples from U2OS cells transfected with a YFP-ALC1 construct using a GFP-trap or an IgG control, as described on the method section. Proteins were resolved in SDS PAGE and blotted with antibodies against ALC1 or eIF4A1. A representative western blot is shown. INPUT samples from cells transfected or not with the YFP-ALC1 are shown on the right side. Endogenous ALC1 and YFP tagged ALC1 bands are marked with arrows. INP. **B** A representative western blot (of three) with protein samples from U2OS cells transfected with siRNAs against ALC1, eIF4A1, or both, or against a control sequence, and bearing a GFP-CtIP or a GFP plasmid. The signals from CtIP, ALC1, eIF4A1 and HSP70 antibodies are shown. **C** Resection in cells treated as in (A) was measured by RPA foci formation at 1 h after irradiation (10 Gy). The average and standard deviations of three independent experiments are shown. Statistical significance was calculated using a two-way ANOVA test. One, two or three asterisks represent P<0.05, P<0.01 and P<0.001 respectively. **D** A representative image of each condition shown

in (C). Scale Bar represent 25 μm. **E** Homologous recombination was measured as described in Fig 1A but using the DR-GFP reporter (left) in cells snRNA-depleted for only eIF4A1 or for both eIF4A1 and ALC1. The average and standard deviations of three independent experiments are plotted. Statistical significance was calculated using a one-way ANOVA test. Three asterisks represent p<0.001.

whole protein were not studied. Importantly, the G4 5′-UTR-transcript of *CtIP* is the most abundant transcript of those that code for the full-length protein in the cells, accounting for over two thirds of the *CtIP* mRNA that codes for a wildtype protein in normal conditions (Fig 6B). Moreover, only the G4 variant was reduced upon ALC1 depletion (Fig 6B). Similar results were obtained in HeLa cells, reinforcing the idea that this is a general response in different cell lines (Fig 6C). CtIP is known to be heavily regulated during the cell cycle [7–9,37,38]. In order to see if any of those transcripts was particularly affected by ALC1 in a specific cell cycle stage, we synchronized RPE1 cells in G0/G1 with serum starvation and measured the presence of each CtIP isoform 0 h after fresh serum addition (G1 enriched cells) or 22 h later (samples enriched in S/G2 cells) (S5D fig). As expected, both transcripts were more abundant in the samples enriched for S and G2 cells (Fig 6D and 6E), in agreement with CtIP expression been cell cycle regulated. Whereas the G4less mRNA was not affected in any cell cycle phase (Fig 6D), the G4-bearing transcript was similarly affected regardless of the cell cycle stage (Fig 6E).

As only one of the CtIP isoforms was affected, together with the fact that the ATPase activity of ALC1 was not required for DNA end resection, we hypothesized that this regulation of mRNA levels might happens post-transcriptionally, probably at the level of RNA stability. To test this idea, we analyzed how much mRNA of both CtIP isoforms decay after transcription inhibition with actinomycin D (ActD) for 12 h compared with DMSO treated cells as a control and normalized to the G4less transcript in cells transfected with the control siRNA (Fig 6F). As expected, the G4 transcript was expressed to a higher degree than the G4less in all cases (Fig 6F). Similar amounts of G4less mRNA were observed in DMSO and ActD treatment in cells transfected with control siRNA, indicating that such transcript is quite stable (Fig 6F). On the contrary, a 25% decrease of the G4 mRNA levels was observed upon treatment with ActD on control cells, suggesting that such isoform turnover is indeed faster than the G4less, but still quite stable (Fig 6F). Thus, the differences in abundance between G4less and G4 transcripts cannot be ascribed to changes in stability per se. Strikingly, ALC1 and eIF4A1 depletion increases the decay of both species of *CtIP* mRNA, as the amount of RNA was greatly reduced in ActD treated samples when compared with DMSO. However, for ALC1 this effect was more evident in the G4 5′-UTR mRNA and only to a lesser extent the G4less 5′-UTR molecule (Fig 6C). Indeed, in ALC1 depleted cells the G4 5′-UTR mRNA was halved in ActD treated versus DMSO cells, but the G4less 5′-UTR was decreased only around 33%. In stark contrast, eIF4A1 downregulated cells showed a decrease of 75% in both *CtIP* splicing variants. Therefore, our data indicate a role of both ALC1 and eIF4A1 in controlling the stability of *CtIP* mRNA, but especially when the G4 5′-UTR was present. In summary, the G4 transcript accumulates to a higher level than the G4less in control cells, in a fashion that does not depend on mRNA stability. Additionally, ALC1 and eIF4A1 are especially important to maintain the G4, and only to a lesser extent the G4less, mRNA stability.

## The presence of the G4 5′-UTR sequence of CtIP is sufficient to render the stability of GFP mRNA as dependent on ALC1 and eIF4A1

To unequivocally demonstrate that the G4 5′-UTR of *CtIP* mRNA was responsible for the requirement of ALC1 and/or eIF4A1 for full CtIP expression, we cloned it upstream of a GFP

**A**

**"G4less" (Transcript ID: ENST00000399722.6)**

TGCGCCGACTGCGGCTCGCGCGCGCGCTTCGGAGGTTTTTTGGCCAGACCCGCACGCGGAACCGGCGCGGGCACCTGGGGGAGAAAT
GGATGGAGAAGGGACCTGGCTGGAAAGCCTTTGCCCCGCTGCTCTGCTCCGCCCATAAGAGGACCCCTGAAATGTCCCGTGCAGTTTG
TTCAAGTCCCCTGTGTGATGAAATGTGCCTCTCGCCTTACCCGTGTGAGAATACCTGTGGTGTGGCAGCGAGTATTTTGG*TATTTGACCTG
TCCAAAGACGACTTGATACCTCTATAATGTAACAGAAAAGGTCAGAAAATATTAAGCAAGTAGAAGTGTGGAGCATATTAAGCAAG*

**"G4" (Transcript ID: ENST00000327155.9)**

AAGTGGAACTCCCGCGTGACGTCGCGCGGGCTCCCGGGCGGGGCGGGTCCGGCCGCCTCCGAGCCCGGCCGGCAGCCCCCGGCCT
TAAAGCGCGGGCTGTCCGGAGGGGTCGGCTTTCCCACCGAGGATTTGGCACTCTGGTGAGGGAAAAGGGCGAAAGAGAAAAGCGAGC
AGCCGTCCTTTCACAGCCTCAGAAAGTGCTCGCTTCCCTTCGGGGGCTTTCGCGAATCCCGAGGCAATCTCGGAGGCGG*TATTTGACCT
GTCCAAAGACGACTTGATACCTCTATAATGTAACAGAAAAGGTCAGAAAATATTAAGCAAGTAGAAGTGTGGAGCATATTAAGCAAG*

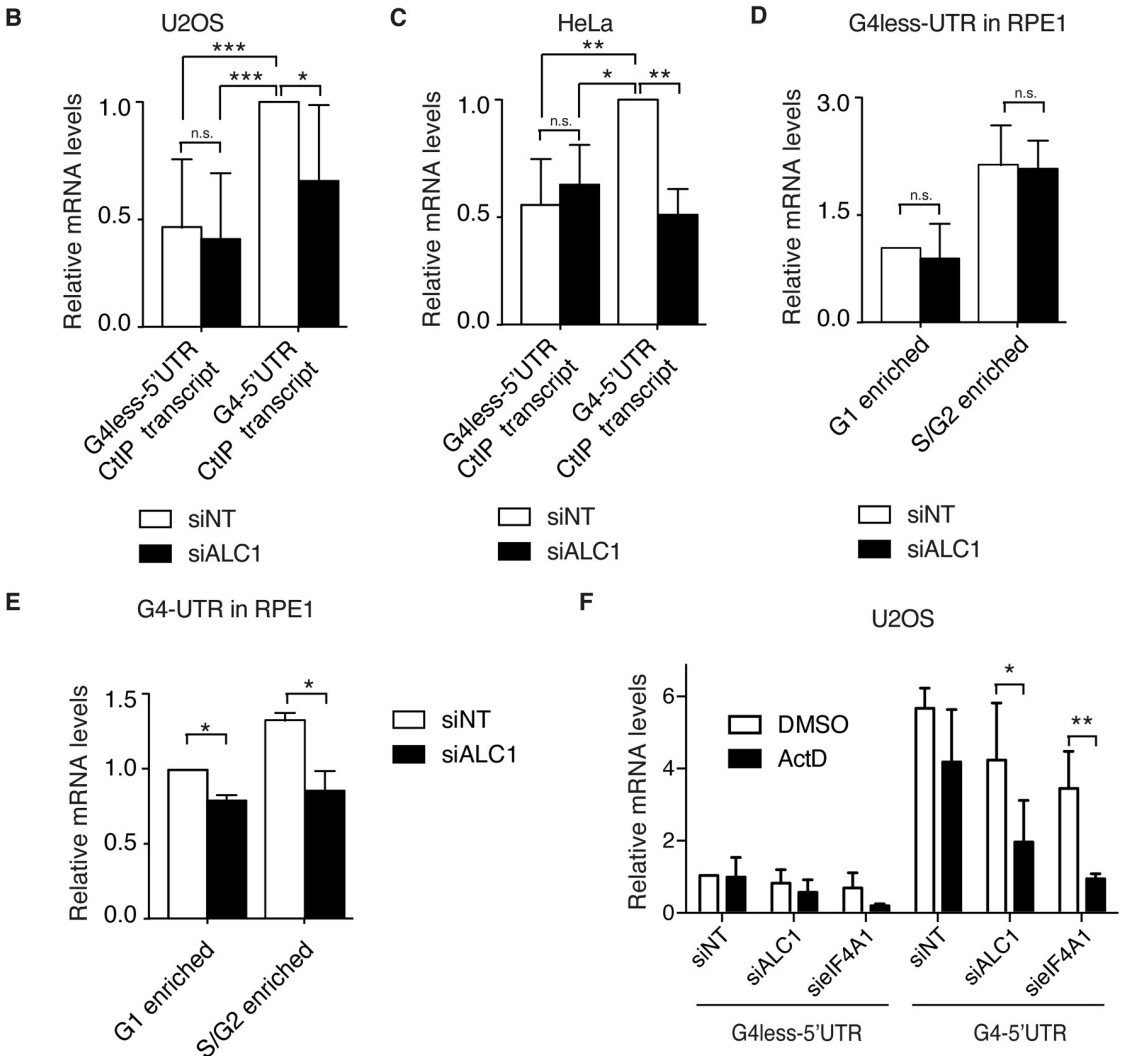

**Fig 6. The 5′-UTR sequence of *CtIP* mRNA determines its requirement for ALC1 and eIF4A1. A** 5′-UTR sequences of the two splicing variants of CtIP that code for a full-length protein. The common sequence is labelled in blue. CGG repeats, typical of eIF4A1-binding sites on mRNA, are shown in red. The transcript that is not likely to form RNA G-quadruplexes (G4less is shown at the top, and the one that is likely to form G4s (G4), as recognized by QGRS Mapper software, is shown at the bottom. The sequence prone to forming the G-quadruplex in the G4 variant is depicted as green boxes. **B** qPCR using splicing-variant specific primers of the G4 and G4less *CtIP* mRNA in U2OS cells depleted for ALC1 (black bars) or not (white bars). All mRNA data were normalized to the G4 isoform of *CtIP* in the control conditions. The average and standard deviation of three experiment is shown. Statistical significance was calculated using a two-way ANOVA test. One, two or three asterisks represent P<0.05, P<0.01 and P<0.001 respectively. **C** Same as panel B, but in HeLa cells transfected with the indicated siRNAs. **D** qPCR using specific primers of the G4less *CtIP* mRNA in RPE cells depleted for ALC1 (black bars) or not (white bars) 0h (G1 enriched cells) or 22h (S/G2 cells enriched samples) after serum starvation and release. All mRNA data were normalized to the G1 enriched sample in the control conditions. Other details as in B. **E** Same as panel D, but using primers specific for the G4 isoform. **F** Accumulation of the G4+ and G4− variants of *CtIP* mRNA in cells depleted for the indicated proteins after 12 h of treatment with actinomycin D (ActD) or DMSO (as a control). The other details are as given in (B).

gene (G4-GFP). As controls, we used the unmodified GFP and another construct containing the 5′-UTR of the CtIP mRNA transcript that lacks the G4 sequence (G4less-GFP). GFP was equally expressed in all genetic backgrounds in cells transfected with the GFP plasmid control (Fig 7A). Interestingly, GFP accumulation from the G4-GFP construct was higher than that from the G4less-GFP plasmid (Fig 7B), mimicking what happens with both endogenous isoforms of full-length *CtIP* mRNA (Fig 6B) and suggesting that such a UTR on its own might increase the mRNA and protein amounts. More importantly, GFP protein levels depended on the presence of both ALC1 and eIF4A1 in cells transfected with the G4-GFP but not from cells transfected with the G4less-GFP construct (e.g., eIF4A1 or ALC1 depletion had no effect on GFP levels in G4less-GFP cells) (Fig 7B). We then confirmed that the protein levels reflected changes in the accumulation of mRNA. Again, there was little effect on a standard GFP plasmid control (Fig 7C), but strikingly, *G4less-GFP* mRNA accumulation was independent of

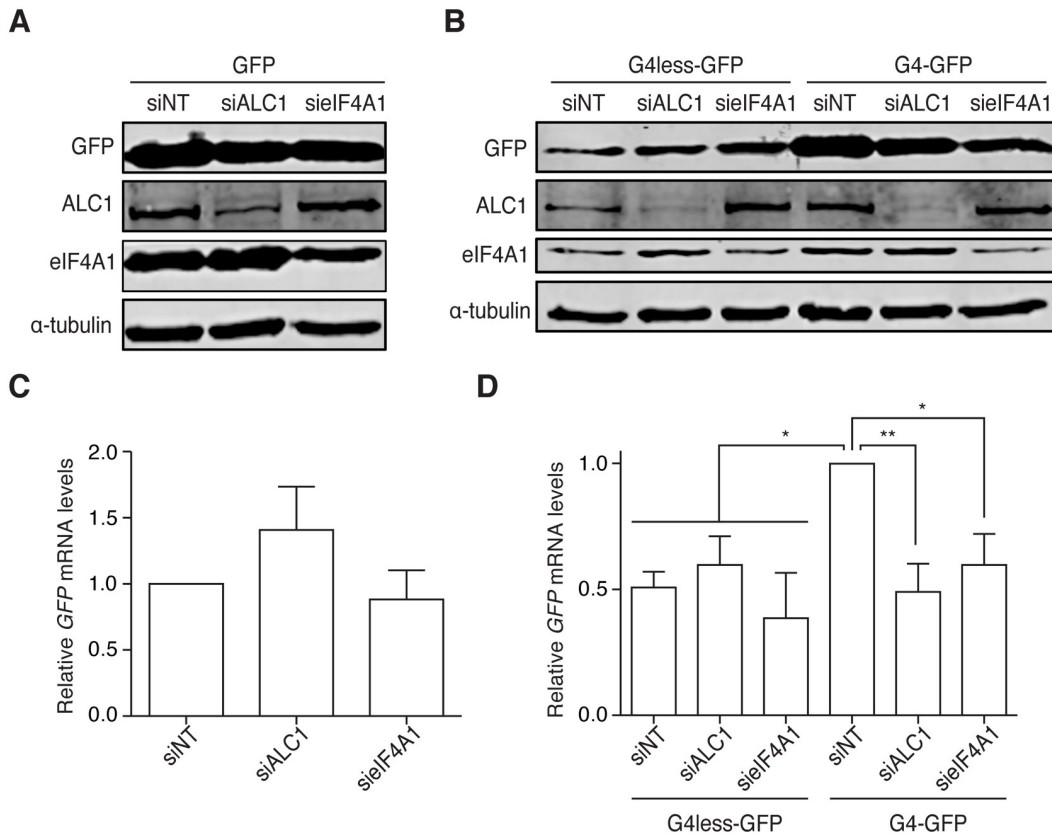

**Fig 7. The G4 5′-UTR sequence of *CtIP* renders any mRNA dependent on ALC1 and eIF4A1 for stability. A** Representative western blot (of three) showing the accumulation of GFP from a control GFP plasmid in cells transfected with the indicated siRNAs. Briefly, cells transfected with a GFP construct were depleted of ALC1 or eIF4A1. Protein samples were collected and resolved in an SDS PAGE before blotting with the indicated antibodies. **B** RNA was isolated from U2OS cells transfected with an empty GFP control plasmid and depleted for the indicated factors. The amount of *GFP* mRNA was measured using qPCR and normalized to the samples transfected with the control siRNA. The average and standard deviation of three independent experiments are plotted. Statistical significance was calculated using a one-way ANOVA test. **C** The same as in (A) but in cells with constructs in which the *GFP* gene was fused to either the G4⁻ (left) or G4⁺ 5′-UTR of *CtIP*. **D** The same as in (B) but in cells transfected with the transcription fusions of CtIP 5′-UTRs and GFP. The average and standard deviations of four independent experiments are shown. Statistical significance was determined by a one-way ANOVA test. Statistical significance was calculated using a one-way ANOVA test. One or two P<0.05 and P<0.01, respectively.

ALC1 and eIF4A1, whereas *G4-GFP* mRNA required both proteins for proper accumulation ([Fig 7D]). Thus, the presence of the G4 from CtIP in the unrelated 5′-UTR of the *GFP* gene is sufficient to make the expression of that gene dependent on ALC1 and eIF4A1.

## Discussion

The balance between HR and NHEJ is critical for cell fitness and an appropriate response to chromosome breaks. Thus, it is not surprising that a tight regulatory network controls the choices between the two DSB repair pathways. In the last decade, it has become obvious that the preferred regulatory point is DNA end resection, and more specifically, the control of the activity and levels of the key resection factor CtIP. CtIP is heavily post-transcriptionally modified, allowing it to specifically respond to cell cycle positioning and the DNA damage response [6]. Constitutive post-transcriptional modifications also impact in its role in resection [14]. However, it is also heavily controlled both transcriptionally and at the level of protein stability [7–12]. Here, we show an additional layer of regulation of CtIP homeostasis, which acts post-transcriptionally at the level of mRNA stability. Of the two known splicing variants that code for a full-length protein, we show that the most abundant form (the mRNA of which contains the G4 structure; termed here G4) is intrinsically less stable. Moreover, both variants (but mostly the most abundant one) require additional factors that control their stability, namely the helicases ALC1 and the transcription factor eIF4A1. ALC1 depletion has little effect on the bulk of *CtIP* mRNA, as detected using a pair of oligonucleotides that recognize all *CtIP* isoforms and not only the two that code for full-length CtIP; this argues that it does not affect general *CtIP* transcription. But a different picture emerges with a careful analysis specifically of the two splicing variants that code for a fully functional CtIP. In this case, changes in mRNA levels can be readily observed in the G4 variant. Thus, although ALC1 has little effect on the bulk of *CtIP* splicing forms, it is extremely important for maintaining the levels of the full-length CtIP protein, which it does by affecting the stability of the mRNA variants that encode the full-length protein and, particularly, for the G4 isoform. Most of this depends on the presence of specific 5′-UTR sequences. Indeed, of the two isoforms analyzed here, termed G4 and G4less, the one containing RNA sequences that are prone to forming G-quadruplexes (G4) is especially relevant. The presence of the G4 sequence, even in a chimeric fusion with GFP, increases the levels of mRNA and protein as compared with the sequence of G4less, but, on the negative side, it makes such splicing variant more dependent on ALC1. Interestingly, this is more evident in the context of the GFP constructs than in the endogenous *CtIP* mRNA, particularly for the requirement of eIF4A1. One likely explanation is that the *CtIP* mRNA contains additional signals that control its stability, which could be dependent as well as independent of ALC1 and eIF4A1. The analysis of the GFP construct, at the level of mRNA and protein accumulation, also indicates that, superseded to the regulation of mRNA stability, an additional layer of control seems to exist at the level of protein translation that relies on eIF4A1 but not on ALC1. Indeed, siRNA-mediated depletion of ALC1 and eiF4A1 rendered similar effects on the mRNA amount from the GFP constructs but showed dramatic differences at the protein level. Specifically, whereas downregulation of eIF4A1 increased GFP levels from a G4less-harboring mRNA, it severely reduced protein production from the G4 construct as compared with cells depleted for ALC1 and despite having similar mRNA levels. Moreover, even though both GFP constructs are severely decreased at the level of mRNA after eIF4A1 depletion, a reduction in protein accumulation is only evident in the G4 GFP-fusion. As mentioned above, both isoforms contain the (CGG) motif that can be recognized by eiF4A1, but only the G4 variant can form a secondary structure of the RNA G-quadruplex. Thus, it is not surprising that the

stability of both mRNA species can depend on eIF4A1, but the translation of only the G4-containing one is affected. This agrees with the known role of eIF4A1 in translation of mRNAs that contain G-quadruplexes in their 5′-UTR [35].

However, and as mentioned above, these results are less clear in the context of the endogenous *CtIP* mRNA splice variants, suggesting additional effects of ALC1, eIF4A1 and possibly other proteins in controlling *CtIP* mRNA stability and maybe translation efficiency. In fact, the splicing factors SF3B and Aquarius were recently shown to maintain CtIP levels and proper HR rates [10,39]. Interestingly, SF3B affect resection mainly through CtIP, but as we observe here with ALC1, SF3B effect on HR cannot be rescued by CtIP overexpression [39]. Moreover, the mRNA binding protein SERBP1, associated to the 40S subunit of active ribosomes in normal and tumor cells [40], binds to *CtIP* mRNA and promotes its translation specifically in S phase by binding to its 3′-UTR [41]. These additional factors might explain some discrepancies observed between ALC1 and CtIP depletion for DNA end resection. Chiefly among them, ALC1-depleted cells are not sensitive to camptothecin treatment [20], but CtIP has been shown to be particularly important for CPT-induced DSB resection and survival [42]. CPT adducts only become DNA DSBs in S phase associated to replication. In contrast, IR or etoposide promote DSB formation in all of the cell cycle stages. Therefore, ALC1 may have a more important role in maintaining CtIP levels in G2, whereas it could be counteracted in S phase by SERBP1 [41]. One likely explanation is that in S phase, SERBP1-induced increases in translation might outweigh the reduction of *CtIP* mRNA levels caused by ALC1 depletion, allowing a normal expression of the protein and sufficient resection of DSB in S phase. This additional layer of regulation of CtIP expression at the level of mRNA stability/translation probably reflects its importance for replication. Indeed, CtIP is known to be enriched in ongoing replication forks [43], and to be important for rescuing stalled replication forks [44]. Further, CtIP protects the forks from degradation when a replication stress is present [45]. Thus, another possibility is that, despite the reduced levels of CtIP after ALC1 depletion, all the remaining protein remains associated with the replication machinery, where it will be readily available to repair CPT-induced damage, but it is not free to support resection outside of the replication context.

Our data also indicate that ALC1 has an additional role in controlling the response to DNA breaks that is independent from its DDR function of relaxing chromatin in the vicinity of DSBs [21]. This new role relies in controlling CtIP to regulate resection. Whether this novel activity is connected with the DNA damage response remains to be explored. Additionally, our data also suggest that ALC1 affects HR independently of CtIP levels, and this effect could rely in the regulation of the levels of other HR proteins or in the role in the DDR previously described [20,21]. One interesting hypothesis is that the G4less variant serves as a reservoir of *CtIP* mRNA, in order to grant the production of a basal level of the protein that is less affected by external signals. In contrast, the G4 5′-UTR specie is responsible for the bulk of protein expression but might be more sensitive to the presence of specific factors. One tantalizing idea is that this provides for an alternative regulatory step in DNA end resection and that, in fact, the G4 5′-UTR isoform abundance might react to yet unknown signals. ALC1 and eIF4A1 might participate in this unknown regulation. In fact, it is plausible that ALC1 retention at sites of DSB might modulate its ability to stabilize *CtIP* mRNA. Usually, ALC1 is quickly recruited to DNA damage, but it is also quickly released [20]. Thus, normally CtIP expression would not be affected. However, in this scenario, trapping ALC1 at damaged chromatin (for example, because the breaks cannot be repaired) could destabilize the mRNA, causing a drop in the CtIP protein level in the long term, and thus preventing hyper-resection.

## Supporting information

**S1 Table. siRNAs used in this study.**
(DOCX)

**S2 Table. Primary antibodies used in this study.** WB: Western blot; IF; Immunofluorescence;
SMART: Single Molecule Analysis of Resection Tracks.
(DOCX)

**S3 Table. Secondary antibodies used in this study.** WB: Western blot; IF; Immunofluorescence; SMART: Single Molecule Analysis of Resection Tracks.
(DOCX)

**S4 Table. Primers used in this study.**
(DOCX)

**S1 Fig. ALC1 depletion. A**, Gene conversion, measured as described in Fig 1A in cells
depleted of ALC1 using two additional siRNAs. siALC1-2 targets the coding region, whereas
siALC1- 3' UTR targets the 3' UTR of the RNA. **B**, Same as A, but using the EJ5 reproter to
check NHEJ efficiency. **C,** Depletion efficiency of ALC1 in cells transfected with the indictaed
siRNAs. **D**, Same as A, but using additional siRNA against ALC1, either ALC1-2 (left) or an
siRNA targeting the 3' UTR (right) in U2OS cells. **E,** Cell cycle profile of U2OS cells transfected either with siNT control sequence or with siRNAs against ALC1 and CtIP as indicated.
The average and standard deviation of three independent experiment are plotted. **F,** Same as C
but in HeLa cells. **G**, The average number of RPA foci per cell formed 1h upon exposure to
10Gy of ionizing radiation was scored automatically using the software Metamorph. The average and standard deviation of three independent experiments are plotted (left) and representative images of one cell are shown on the right. Statistical analysis as described in Fig 2A.
(PDF)

**S2 Fig. Accumulation of anti-resection factors in ALC1 depleted cells. A**, The average number of 53BP1 foci per cell was calculated as described in Fig 3A in cells at different times after
exposure to 2 Gy of ionizing radiation in control cells and cells depleted for ALC1 (red). **B**,
Computer-based measurement of the size of 53BP1 foci in cells depleted for the indicated factors using the Metamorph software. Size was normalized with the control sample. The average
and standard deviation of three independent experiments is plotted. Other details as Fig 1A. **C**,
Computer-based analysis of the intensity of 53BP1 foci. Other details as panel B. **D**, Same as B
but for RIF1 foci. **E**, Same as C but for RIF1. **F**, Same as B but for CCAR2 foci. **G**, Same as C
but for CCAR2.
(PDF)

**S3 Fig. Resection impairment in ALC1 depleted cells does not depend on the loading of
anti-resection factors. A,** Resection was measured 1h after irradiation in cells infected with
shRNA against 53BP1, CCAR2 or a control sequence and transfected with an siRNA against
either ALC1 (black bars) or a control sequence (white bars). Representative images are shown
on the right. Scale Bar represent 25 μm. Other details as in Fig 5B. **B**, Western blot showing the
downregulation of ALC1, CCAR2 and 53BP1 of the cells described in A. HSP70 was used as
loading control. **C**, Cell cycle analysis of the cells described in panel A.
(PDF)

**S4 Fig. Resection impairment in ALC1 depleted cells is suppressed by different ALC1 constructs. A,** Resection was measured 1h after irradiation in cells transfected with siRNA against
ALC1 (black bars) or a control sequence (white bars) and then transfected with the different

YFP-ALC1 constructs as mentioned in Fig 2. Representative images are shown on the right. Scale Bar represent 25 μm. Other details as in Fig 2A. **B**, The average number of RPA foci per cell in samples treated as in panel A and formed 1h upon exposure to 10Gy of ionizing radiation was scored automatically using the software Metamorph. The average and standard deviation of three independent experiments are plotted. **C**, Cell cycle analysis of the cells described in panel A. **D**, Western blot showing the expression of YFP-tagged version of ALC1. Protein samples from cells transfected with the indicated ALC1 variants, downregulated or not for the endogenous version using siRNA as depicted in the figures, were resolved in SDS-PAGE and blotted with an ALC1 antibody. The YFP-tagged and endogenous proteins are marked in the blot with triangles. α-tubulin was used as loading control. **E**, The levels of CtIP protein in cells transfected with the mentioned version of YFP-ALC1 and depleted (black bars) or not (white bars) of endogenous ALC1 with an siRNA targeting the 3' end of the mRNA was determined by Western blot quantification using the Odissey Li-Cor Infrared system and normalized to control cells transfected with an empty plasmid. The average and standard deviation of three independent experiment are plotted.
(PDF)

**S5 Fig. Cell cycle analysis of ALC1 and eIF4A1 depleted cells bearing GFP and GFP-CtIP. A**, The levels of CtIP protein in cells depleted for the indicated proteins was determined by western blot quantification using the Odissey Li-Cor Infrared system and normalized to control cells. A representative image is shown on the right side and the average and standard deviation of three independent experiment are plotted on the left. Statistical significance was calculated using a one-way ANOVA test. One, two or three asterisks represent $p<0.05$, $p<0.01$ and $p<0.001$, respectively. **B**, Cell cycle analysis of the cells described in Fig 5B. **C**, Gene conversion was measured as described in Fig 1A in cells bearing with a FLAG-CtIP or an empty FLAG plasmid and transfected with the indicated siRNAs, **D**, RPE1 cells synchronized by serum starvation and release as described in the methods section were collected 0h or 22h after serum addition and cell cycle distribution was assayed by FACs analysis. At 0h an enrichment in G1 cells was observed, whereas 22 h later an enrichment on S and G2 was detected.
(PDF)

## Acknowledgments

We wish to thank Simon Boulton for kindly providing YFP-ALC1 variants, and Nestor García-Rodríguez and Sabrina Rivero for critical reading of the manuscript.

## Author Contributions

**Funding acquisition:** Pablo Huertas.

**Investigation:** Fernando Mejías-Navarro, Guillermo Rodríguez-Real, Javier Ramón, Rosa Camarillo.

**Supervision:** Pablo Huertas.

**Writing – original draft:** Pablo Huertas.

**Writing – review & editing:** Pablo Huertas.

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
