## [Decision Letter · Decision Letter 0]

23 Oct 2019

Dear Dr Huertas,

Thank you very much for submitting your Research Article entitled 'ALC1/eIF4A1-mediated regulation of CtIP mRNA stability controls DNA end resection' to PLOS Genetics. Your manuscript was fully evaluated at the editorial level and by independent peer reviewers. The reviewers appreciated the attention to an important and interesting problem, and appreciated your discovery of an unanticipated mechanism of regulation.  They also raised some substantial concerns about the current manuscript which need to be addressed. Based on the reviews, we will not be able to accept this version of the manuscript, but we would be very happy to receive a revised version for re-review.

If you decide to revise the manuscript for further consideration at PLOS Genetics, please aim to resubmit within the next 60 days, unless it will take extra time to address the concerns of the reviewers, in which case we would appreciate an expected resubmission date by email to plosgenetics@plos.org.

[LINK]

We are sorry that we cannot be more positive about your manuscript at this stage. Please do not hesitate to contact us if you have any concerns or questions.

Yours sincerely,

Nancy Maizels, Ph.D.

Associate Editor

PLOS Genetics

Gregory P. Copenhaver

Editor-in-Chief

PLOS Genetics

Reviewer's Responses to Questions

**Comments to the Authors:**

Reviewer #1: The manuscript by Mejias-Navarro identifies a novel mechanism that controls DNA end resection by regulating the expression of a key resection factor CtIP. It is demonstrated that ALC1 has a structural role, together with the transcription factor eIF4A1, to regulate the stability of the major mRNA coding for CtIP. Specifically, the depletion of either ALC1 or eIF4A1 decreases the stability of CtIP mRNA. Interestingly, the effect is linked to a 5' UTR of mRNA CtIP, and, elegantly, it is demonstrated that fusing this UTR (containing a G-rich sequence) to an unrelated mRNA results in the same regulatory effect.

The works is generally well-done. One conceptual point that is missing is a lack of understanding when and where such a regulation would take place. Dr. Huertas earlier showed that Sae2 and CtIP are regulated by post-translational manner by phosphorylation, which links cell cycle regulation to resection, and which generally allows resection when sister chromatids are available. Furthermore, resection is further e.g. regulated by ATM phosphorylation of CtIP, which responds to the presence of DNA damage. Can the authors comment on what function the mechanism involving ALC1/eIF4A1 has? Is it cell cycle or DNA damage regulated?

Furthermore, while I am in most cases convinced by the data presented, one exception is the quantitation of RPA positive cells. Looking at the representative images shown, it seems to be very difficult to decide whether a cell is "positive" or "negative", as there appears to be a whole range of RPA intensities from almost invisible to very strong. It would be more convincing to show FACS profiles for some of the key experiments (Fig. 1 and Fig. 4).

The authors conclude that the "5' UTR of CtIP is sufficient to render the stability of any mRNA", while in fact only one other RNA was tested. It would certainly be interesting to perform transcriptome-wide analysis upon ALC1/eIF4A1 depletion and correlation with the likelihood of G4 being formed at UTR; otherwise, the authors should rephrase the statement.

Below are minor points and typos I found while the reading the text

page 3: "this include the activation"

page 9: Fig 1A and B, panels A and B

page 10: Fig Supplementary Fig

page 10: We observed a reduced not only in the number

page 12: Fig. 5A does not show double-depletion

page 13: CtIP gene produce two different

Reviewer #2: The study from Mejías-Navarro and colleagues report that loss of ALC1 (also called CHDL1) reduces homologous recombination (HR), which they attribute to reduced end-resection measured by RPA foci upon DNA damage. In this context, they find that ALC1 knockdown reduces the protein level of CtIP, a factor implicated in the initiation of DNA end-resection. Surprisingly, their data suggest that ALC1 regulates CtIP level independently from its enzymatic activity or PAR-binding. They confirm the interaction between ALC1 and eIF4A1 that previously identified in proteomic screens. It is known that eIF4A1 is an RNA helicase implicated in the resolution of G4 RNA structure. Using bioinformatic tools, they identify 2 isoforms of CtIP mRNA with different 5’UTR sequences due to alternative splicing. They suggest that ALC1 and eIF4A1 regulate the stability of the CtIP isoform with the G4 structure. Analyses of the GFP with G4 or G4less 5’UTR derived from CtIP confirm the findings. Overall, the study identified a new mean to regulate end-resection, by modulating the stability of CtIP mRNA through ALC1 and eIF4A1. The strengths are the novel mechanism and the potential chromatin independent role of ALC1 in DNA repair. CtIP protein levels are low in G1 and much higher in S/G2. It is important to clarify whether the different isoforms of CtIP transcripts present in different cell cycle phases and whether ALC1 and eIF4A1 have cell cycle independent roles on CtIP level regulation. Meanwhile, whether the G4-mediated regulation of CtIP contributes to the cell cycle phase dependent regulation of CtIP protein level. In addition, there are a few other concerns limited the enthusiasm to the manuscript.

Major concerns:

1) The observations on the un-coordinate changes between CtIP mRNA level and protein level upon ALC1 or eIF4A1 depletion are quite interesting. This is also true for cell cycle specific regulation of CtIP protein levels. Sup Figure 5B showed that eIF4A1 knockdown caused significant G1 and G2/M arrest in the cells test. It is important to understand whether eIF4A1 can directly regulate CtIP stability independent of the cell cycle phases.

2) Can CtIP expression rescue the HR defects in ALC1 deficient cells? This experiment is important to establish CtIP reduction as the main mechanism by which ALC1 contributes to HR.

3) In figure 2D, the impact of ALC1 knockdown has a much more dramatic effect on RPA foci formation in telomerase positive Hela cells, than ALT- U2OS cells. Does the two form of CtIP transcripts also exist in Hela cells and are they also protected from ALC1 and eIF4A?

4) Figure 5C showed that ectopic expression of CtIP without the 5’UTR rescued the resection measured in siALC1 cells. How about the CtIP cDNA with endogenous 5’-UTR?

Minor comments:

1) Page 4 – line 4 “…… the presence of DSBs triggers a complete, massive cellular response.” Should it be complex, instead of complete?

2) The author should tone down the discussion regarding the chromatin dependent function of ALC1. Although the CtIP mRNA regulation by ACL1 seems independent of it helicase activity or the Macro domain, there is no direct evidence in the manuscript to exclude the possibility that ALC1 might also promote HR through other mechanisms beyond regulating CtIP mRNA.

3) There is some inconsistency on whether CtIP G4-mRNA is more stable or at higher accumulative level. It would be helpful to clarify this when it was first measured.

4) The official gene number of ALC1 is CHDL1. It would be preferred to use CHDL1.

**Have all data underlying the figures and results presented in the manuscript been provided?**

Reviewer #1: None

Reviewer #2: Yes

PLOS authors have the option to publish the peer review history of their article (what does this mean?). If published, this will include your full peer review and any attached files.

Reviewer #1: No

Reviewer #2: No

---

## [Decision Letter · Decision Letter 1]

22 Apr 2020

Dear Dr Huertas,

We are pleased to inform you that your manuscript entitled "ALC1/eIF4A1-mediated regulation of CtIP mRNA stability controls DNA end resection" has been editorially accepted for publication in PLOS Genetics. Congratulations!

Yours sincerely,

Nancy Maizels, Ph.D.

Associate Editor

PLOS Genetics

Gregory P. Copenhaver

Editor-in-Chief

PLOS Genetics

Comments from the reviewers (if applicable):

Reviewer's Responses to Questions

**Comments to the Authors:**

Reviewer #1: I am happy to support the revised version of the manuscript for acceptance.

Reviewer #2: The authors have addressed all my major concerns despite some technical difficulties. In the revision, they have also discussed the results in the context of other findings, including potential CtIP independent role of ALC1/CHD1L in homologous recombination.

**Have all data underlying the figures and results presented in the manuscript been provided?**

Reviewer #1: None

Reviewer #2: Yes

PLOS authors have the option to publish the peer review history of their article (what does this mean?). If published, this will include your full peer review and any attached files.

Reviewer #1: No

Reviewer #2: No

**Data Deposition**

http://datadryad.org/submit?journalID=pgenetics&manu=PGENETICS-D-19-01606R1

**Press Queries**

---

## [Editor Report · Acceptance letter]

4 May 2020

PGENETICS-D-19-01606R1 

ALC1/eIF4A1-mediated regulation of *CtIP* mRNA stability controls DNA end resection 

Dear Dr Huertas, 

We are pleased to inform you that your manuscript entitled "ALC1/eIF4A1-mediated regulation of *CtIP* mRNA stability controls DNA end resection" has been formally accepted for publication in PLOS Genetics! Your manuscript is now with our production department and you will be notified of the publication date in due course.

With kind regards,

Jason Norris

PLOS Genetics

On behalf of:
